# 1 × 2 Graphene Surface Plasmon Waveguide Beam Splitter Based on Self-Imaging

**DOI:** 10.3390/nano14181538

**Published:** 2024-09-22

**Authors:** Liu Lu, Peng Xu, Liang Zhang, Jia Le, Daifen Chen

**Affiliations:** 1School of Mechanical Engineering, Jiangsu University, Zhenjiang 212013, China; 2212103083@stmail.ujs.edu.cn (P.X.); 18252587621@163.com (L.Z.); le220307@163.com (J.L.); 2School of Energy and Power, Jiangsu University of Science and Technology, Zhenjiang 212003, China; daifen01@163.com

**Keywords:** surface plasmons, waveguides, beam splitters, self-imaging

## Abstract

Based on the principle of self-imaging, a 1 × 2 graphene waveguide beam splitter is proposed in this work, which can split the graphene surface plasmons excited by far-infrared light. The multimode interference process in the graphene waveguide is analyzed by guided-mode propagation analysis (MPA), and then the imaging position is calculated. The simulation results show that the incident beam can be obviously divided into two parts by the self-imaging of the graphene surface plasmon. In addition, the influences of the excited light wavelength, Fermi level, dielectric environment on the transmission efficiency are studied, which provide a reference for the research of graphene waveguide related devices.

## 1. Introduction

Plasmons describe collective oscillations of electrons. They have a fundamental role to excite and regulate the optical signal in the sub-wavelength scale [1,2]. Plasmons of two-dimensional massless electrons, as present in graphene, are some of the most fascinating and potentially new plasmonic metamaterials [3,4,5]. Graphene surface plasmon (GSP)-based technologies will enable the development of compactly and inexpensively active photonic elements because graphene plasmons can be tuned by gratings, dopes, chemical means and through conventional plasmonics based on noble metals. Various graphene-incorporated plasmonic devices have been proposed such as tunable graphene array filters [6,7,8], Bragg reflectors [9,10] and waveguides [11,12,13,14].

A beam splitter is an important part of integrated optics. At present, a variety of methods have been developed to fabricate beam splitters. Metamaterial design is one of the most frequently used methods. By controlling the shapes, periods, sizes and other parameters of the metamaterial units, the reflectivity of the beams can be controlled, so as to achieve the purpose of beam splitting [15,16,17,18]. In reference [19], a T-shaped structure is proposed with a Yttrium Iron Garnet (YIG) rod placed at the bifurcation point. The vibration’s direction of the local plasmons can be changed by the external magnetic fields, thus the beam splitting of the plasmon energy can be realized. In addition, photonic crystals [20,21,22,23] and dielectric gratings [24,25] can also be used to make beam splitters in integrated optical paths. Compared with traditional metamaterials, graphene has a simpler structure and better modulation ability. For instance, a Y-branch graphene waveguide was proposed, in which a monolayer graphene was placed on silicon dioxide substrate with a Y letter-shape [26]. Here, the conductive graphene plasmon mode is bound to transmit inside and divided into two beams at the point of bifurcation. An alternative approach is proposed in reference [12] to add bias voltages to different graphene bands, so that the beam can be truncated or shunted due to the change of the Fermi levels.

In this paper, a 1 × 2 graphene waveguide beam splitter is proposed based on the principle of self-imaging. The theoretical calculation for the process of multimode interferences and the imaging positions are given in the present work. Additionally, the influences of Fermi energy level and the excited light wavelength on the splitter device are also investigated. Furthermore, we also show the numerical simulation result, which is in good agreement with the present theoretical analysis.

## 2. Model Introduction and Theoretical Analysis

It is known that the directional conduction of GSP can be tuned by reasonably constructing local graphene band [27]. Moreover, the number of the plasmonic mode that the graphene waveguide can accommodate depends on the width of graphene [13]. Based on these unique properties, we construct a multimode graphene waveguide, in which the plasmonic modes would interfere in the waveguide. As shown in the schematic diagram in Figure 1, the proposed plasmonic beam splitter is composed of a monolayer (ML) graphene placed between two dielectrics with the relative permittivity $\varepsilon_{1}$ and $\varepsilon_{2}$. This is a typical sandwich structure, which supports the excitation of the GSP. When the refractive index of the graphene is larger than those of the upper and bottom dielectrics, the GSP reflects on the interface between the graphene and the dielectric, resulting in the multimode interference (MMI) of GSP on graphene. If the exit port is set in the imaging position, the GSP energy would be divided into several parts.

In order to study the self-imaging of GSP, the behavior of GSP was numerically analyzed based on the Maxwell equations. For the research model in Figure 1, it is known that the GSP eigenmode satisfying the Maxwell equation is transverse magnetism, and the dispersion relation of TM polarization modes can be expressed as
(1)$$k=\frac{\varepsilon_{0}\left(\varepsilon_{1}+\varepsilon_{2}\right)}{2}\frac{i2\omega }{\sigma }$$
(2)$$n_{\mathrm{eff}}=\frac{k}{k_{0}}$$
where $k_{0}$ is the wavenumber of the free-space and σ(ω) is the conductivity of the graphene. Then, the method of the guided-mode propagation analysis (MPA) is used to calculate the supportive modes [28]. According to the waveguide theory, the propagation constant β of the TM mode of the GSP can be calculated by the following three equations:
(3)$$\tan\left(K\cdot d\right)=\frac{n_{1}^{2}}{n_{2}^{2}}\left(\frac{\gamma }{K}\right)$$
(4)$$K=k_{0}^{2}n_{eff}^{2}-\beta^{2}$$
(5)$$\gamma =\beta^{2}-k_{0}^{2}n_{d}^{2}$$

The parameter $n_{d}$ in Equation (5) is the refractive index of the outer dielectric layer, while d is the width of graphene strip. Here, n_d_ can be defined as the refractive index of the dielectric around the graphene. In general, $\beta^{2}\gg k_{0}^{2}{n_{d}}^{2}$, thus $k_{0}^{2}{n_{d}}^{2}$ could be neglected. After β is obtained, we decompose the input field φ (y, 0) into the weighted superposition of each mode field:(6)$$\Psi \left(y,0\right)=\sum_{v}c_{v}\psi_{v}(y)$$

The field excitation coefficients c can be estimated using overlap integrals:(7)$$c_{v}=\frac{\int \Psi (y,0)\psi_{v}(y)dy}{\sqrt{\int \psi_{y}^{2}(y)dy}}$$

Then, the field profile $\varphi \left(y,z\right)$ at a distance z can be written as a superposition of all the field distributions of the guided modes:(8)$$\varphi \left(y,z\right)=\sum_{v=0}^{m-1}c_{v}\varphi_{v}(y,0)exp[j(\beta_{0}-\beta_{v})z]$$
where m is the maximum number of the modes that can be accommodated in the waveguide. This equation represents the total electric field distribution obtained by superposition of each mode field at the transmission distance of z. The width w of the monolayer graphene in this model is fixed at 3 μm. The dielectric constants of the upper and lower layers are set as 1 and 2.25, respectively, and the thickness of each medium layer is 2 μm. In addition, the width of both the incident and outgoing port is set with 0.5 μm. The incident electromagnetic field propagating along the z-axis (the electric field is TM polarization) is set at z = 0, and the field distribution can be expressed as a Gaussian function for x and y:(9)$$E_{y}=exp(-(x^{2}+y^{2})/{w_{0}}^{2})$$

To get an obvious insight of the self-imaging effect of the waveguide, the parameter of the half width of the Gaussian function $w_{0}$ is set as 0.4 μm. The properties of the graphene can be described using its conductivity σ(ω), which can be given by a simple semi-classical Drude model as the free-space light wavelength is in the infrared range.
(10)$$\sigma \left(w\right)=\frac{e^{2}E_{f}}{\pi \hbar^{2}}\frac{i}{\omega +i\tau^{-1}}$$

Here, the $E_{f}$ is the electronic Fermi energy, which can be adjusted by electrical gate voltage or chemical doping. The parameter τ describes the relaxation time of the charges in graphene, determined by the carrier mobility μ ($\tau =\mu E_{f}/ev_{f}^{2}$). Here, $E_{f}$ and $V_{f}$ are 0.64 eV and 106 m/s, respectively. In addition to the incident boundary with electric field $E_{y}=1$ at z=0, the scattering boundary conditions are applied to the other boundaries of the model. The exit port can be obtained using the MPA theory. We find that the energy of the GSP is divided equally at $z=\frac{3L_{\pi }}{8}$; then, the length of MMI device is defined as $L_{MMI}$, where the energy of the GSP is divided equally. The parameter $L_{\pi }$ is the beat length of the two lowest-order modes:(11)$$L_{\pi }=\frac{\pi }{\beta_{0}-\beta_{1}}≃\frac{4n_{eff}W_{ev}^{2}}{3\lambda }$$
(12)$$W_{ev}≃d+(\frac{\lambda_{0}}{\pi }){\left(\frac{n_{eff}}{n_{d}}\right)}^{2\delta }{\left(n_{eff}^{2}-n_{d}^{2}\right)}^{-\frac{1}{2}}$$

The parameter $W_{ev}$ is the relative width due to the Hangus–Shin displacement phenomenon at the reflection interface of the waveguide.

Moreover, finite element analysis (FEA) modeling and process simulation were carried out in Comsol Multiphysics software. TM polarization incident light was set at the incident port with wavelength of 8 μm. The electric intensity in the x direction was set with Gaussian distribution with a half width of 0.3 μm. A perfect matching layer was set around the structure and at the exit port, while the perfect matching layer was set by scattering boundary condition. The ML graphene can be implemented by a two-dimensional surface current density boundary condition [4]. A free quadrilateral grid distribution was used in the model with a grid size of 50 nm. However, due to the presence of plasmons, the grid size in the graphene is reduced to around 10 nm.

## 3. Results and Discussion

Figure 2a presents a snapshot of the electric field at the x–z cross section under the incident light with the wavelength of 8 μm, while Figure 2b,c present the y component of electric field on the x–z and y–z cross sections, respectively. It is clear that the excited GSP propagates along the surface of the graphene with a wavelength of approximately 295 nm, which agrees well with the theoretical calculation [9]. In addition, we find that the GSP energy is divided into two parts at the two exit ports, as shown in Figure 2b,c. According to the above mentioned MPA theory, the waveguide beam splitter shown in Figure 2 is due to the self-imaging of the GSP, which is different from an all-dielectric waveguide splitter [29,30]. Moreover, we can also observe in Figure 2 that the electric field intensity of GSP decreases with the increasing of the propagation distance, which is due to the inherent loss.

The transmission loss of the graphene plasmons is mainly related to two factors: one is the imaginary part of the effective refractive index $\;n_{eff}\;$ of the graphene surface plasmon, and the larger the imaginary part, the greater the transmission loss; the other is the propagation length $\;L_{MMI}$ of the splitter device. The longer the transmission propagation length, the more energy is lost. In addition, the number of the transmission modes in the waveguide increases with the real part of the effective refractive index $\;n_{eff}$. In addition, because the input electric field is the same, the transmission efficiency can be described by the output fields ${\;\left|E\right|}_{out}$. According to the Equations of (1), (2) and (10), the main factors affecting the parameters of the $n_{eff},\;L_{MMI}$ and ${\left|E\right|}_{out}$ are the Fermi level $E_{f}$, dielectric constant of substrate $\varepsilon_{2}$ and the excited light wavelength λ. Thus, the changes of the $\;n_{eff},\;L_{MMI}$ and ${\left|E\right|}_{out}$ versus the $E_{f}$, $\varepsilon_{2}$ and λ are investigated as shown in Figure 3. As can be seen from Figure 3a,b, when ${\;E}_{f}$ increases, both the $L_{MMI}$ and $Im(n_{eff})$ become smaller, thus both the transmission loss and the propagation length reduce, which results in a consequence that the output electric field ${\left|E\right|}_{out}$ increases. This phenomenon agrees well with the result shown in Figure 3c. In Figure 3c, the dashed line presents the input electric field while the solid lines present the output electric fields under different ${\;E}_{f}$. It demonstrates that with the increasing of the ${\;E}_{f}$ changes, the transmission efficiency increases. Moreover, the cases for the parameters of $\varepsilon_{2}$ and λ were also studied as shown in Figure 3d–f and 3g–i, respectively. We observe that with the increasing of ${\;\varepsilon }_{2}$, both the $L_{MMI}$ and $Im\left(n_{eff}\right)\;$ increase, and accordingly the output electric field intensity declines, as shown in Figure 3f. This means that the transmission efficiency is inversely proportional to the dielectric constant of substrate $\varepsilon_{2}$. However, for the case of the excited light wavelength λ shown in Figure 3g–i, when the λ increases, the $Im\left(n_{eff}\right)\;$ is maintained while the $L_{MMI}$ becomes smaller, and as a result, the output electric field energy reduces. Based on the results shown in Figure 3, compared to the splitter based on dielectric-loaded or metal–insulator–metal surface plasmon polariton waveguides [31,32], we can find that the proposed splitter works at the far-infrared range and the property of the splitter possesses better modulation ability.

In order to quantify the transmission efficiency η  of the GSP waveguide, η  is defined and expressed as $\;\eta =\int {\left|E_{out}\right|}^{2}/{\left|E_{in}\right|}^{2}$. Figure 4 shows η versus the excited light wavelength under the conditions with $E_{f}=0.5\;eV$ and $\varepsilon_{2}=2.25$. To prove our theoretical results, the numerical simulation of the model was carried out using COMSOL Multiphysics and the results are also presented in Figure 4. It is clear that the theoretical values are in good agreement with the numerical simulation results. In detail, when the λ increases from 7 to 12 μm, the transmission efficiency increases from 5% to 40%. However, if the λ keeps increasing, it would result in the reducing of the number of the transmission modes in the graphene waveguide. According to Equation (7), both the input field and the output field can be approximated by the weighted superposition of all modes. When there are few modes, there will be a large difference between the function obtained by superposition and the Gaussian function, and the self-imaging at the exit port will be distorted.

## 4. Conclusions

A plasmonic beam splitter based on self-imaging of the GSP was proposed using the guided-mode propagation theoretical analysis. The influences of the Fermi level, dielectric environment and excited wavelength on the transmission loss of the plasmonic beam splitter were all investigated. It was found that the transmission efficiency of the beam splitter can be effectively controlled by changing the parameters, including Fermi energy level, substrate dielectric constant and excited light wavelength. Moreover, in addition to the theoretical results, we also present the numerical simulation results. The present theoretical results are in good agreement with the simulation results, which verifies the feasibility of using the guided mode analysis method to analyze the plasmon waveguide. Although the proposed plasmonic beam splitter has the disadvantage of unstable output position, multiple parameters can be adjusted to fix the length of the beam splitter. It is believed that the analysis method used in this paper can provide a valuable reference for the research of other GSP devices.

## Figures and Tables

**Figure 1 nanomaterials-14-01538-f001:**
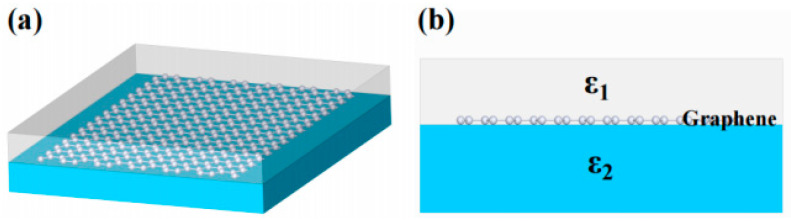
Schematic diagram of (**a**) three-dimensional and (**b**) cross section of the graphene plasmon beam splitter.

**Figure 2 nanomaterials-14-01538-f002:**
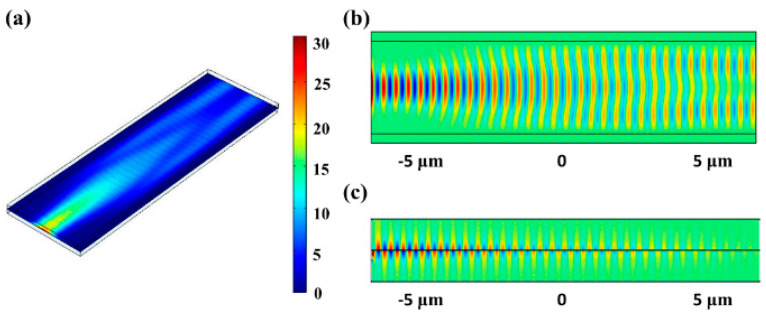
(**a**) The electric field intensity |E| on the graphene surface. The Y component of the electric field on (**b**) the x–z cross section and (**c**) the y–z cross section.

**Figure 3 nanomaterials-14-01538-f003:**
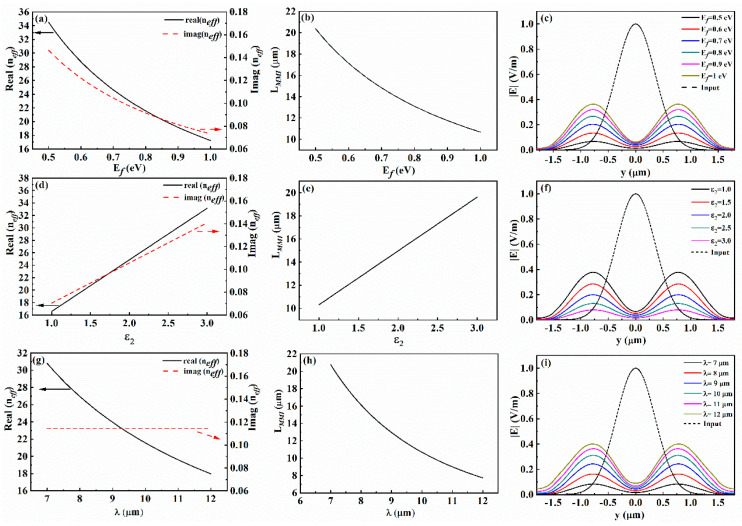
(**a**–**c**) Effective refractive index $n_{eff}$, $L_{MMI}$ and the output field ${\left|E\right|}_{out}$ changed with the Fermi level $E_{f}$, respectively. (**d**–**i**) correspond to the cases of the dielectric constant of substrate $\varepsilon_{2}$ and the excited light wavelength λ, respectively.

**Figure 4 nanomaterials-14-01538-f004:**
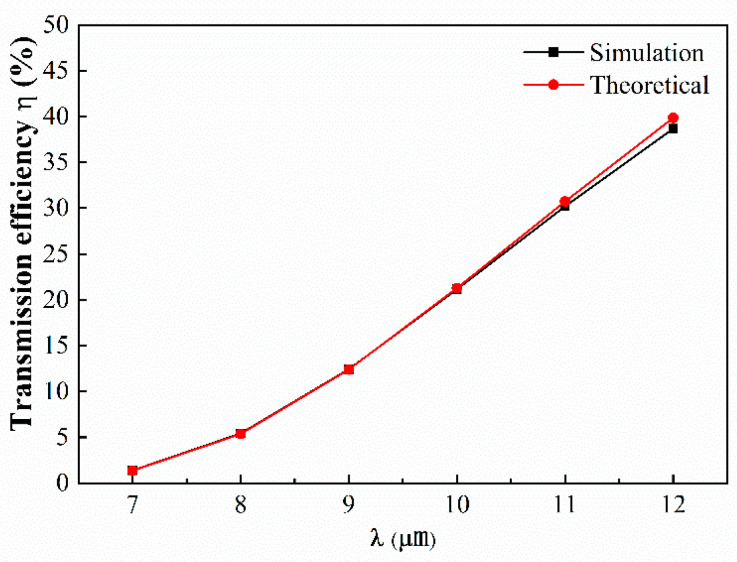
The transmission efficiency η versus the excited light wavelength λ.

## Data Availability

The data presented in this study are available on request from the corresponding author.

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
