# Peer review of "1 × 2 Graphene Surface Plasmon Waveguide Beam Splitter Based on Self-Imaging"

_nanomaterials, 2024, doi:10.3390/nano14181538_

Round 1

Reviewer 1 Report (Previous Reviewer 2)

Comments and Suggestions for Authors

I recommend the article for publication 

Author Response

Comments and Suggestions for Authors
I recommend the article for publication 

Reply: We really appreciate the reviewer’s positive opinion.

Reviewer 2 Report (Previous Reviewer 1)

Comments and Suggestions for Authors

The manuscript has been drastically improved in terms of organization, presentation quality and clarity making it readable and demonstrating thus the technical novelty, which is adequately supported by the theoretical analysis and simulation results.

The manuscript could be accepted for publication in Nanomaterials.

Authors should  consider the following minor issues

1)     The “1*2” should be replaced by “1x2”

2)     The references in brackets [ ]  in lines 54, 55 must be completed

Author Response

Comments and Suggestions for Authors
The manuscript has been drastically improved in terms of organization, presentation quality and clarity making it readable and demonstrating thus the technical novelty, which is adequately supported by the theoretical analysis and simulation results. The manuscript could be accepted for publication in Nanomaterials.

Reply: We really appreciate the reviewer’s positive opinion.

Authors should consider the following minor issues:

1)     The “1*2” should be replaced by “1x2”;

Reply: This is a good suggestion. All the words “1*2” in the paper were replaced by “1x2”, as well as the title.

2)     The references in brackets [ ]  in lines 54, 55 must be completed

Reply: Thanks our reviewer for his attention to detail. The references 1 and 13 were respectively added in the brackets in lines 54 and 55. (See page 1, line 24.)

Reviewer 3 Report (New Reviewer)

Comments and Suggestions for Authors

A graphene monolayer is sandwiched in-between two different media and the structure is used as beam splitter due to the interference of the excited multiple modes. The frequency dispersion of the effective refractive index as well as the influence of the permittivities on the transmission is identified. The transmission efficiency is increasing with the operational wavelength and such a finding, based on analytical modeling, has been also validated via numerical simulations.

The paper is related with an interesting topic but requires several changes before becoming publishable at MDPI Nanomaterials. In particular:

(A) A more professional presentation of both the formulas and the figures is necessary. In addition, most of the figures do not refer to the actual effect but to variations of effective refractive index and prior results.

(B) A meticulous optimization of the reported effect should be executed for various chemical potentials, scattering times of electrons and background permittivity contrast. The regimes for maximum transmission efficiency should be determined.

(C) The authors should discuss extensively in their revised version the possibility of changing the environment into graphene strips are embedded. How the reported effects are expected to change if the surrounding is cylindrical [1] or the whole graphene structure is finite [2]?

(D) A better novelty statement is required. What is the new that this study can offer to related research domain. The authors should elaborate and compare with competing designs.

[1] On examining the influence of a thin dielectric strip posed across the diameter of a penetrable radiating cylinder, Progress In Electromagnetics Research C, 2008.

[2] Excitation of guided waves of grounded dielectric slab by a THz plane wave scattered from finite number of embedded graphene strips: Singular integral equation analysis, IET Microwaves, Antennas & Propagation, 2021.

Author Response

Response: We appreciate our referee for the positive comments. The manuscript has been revised in accordance with the comments. The revised points were highlighted by red color, and our point-by-point responses are as follows:

(A)A more professional presentation of both the formulas and the figures is necessary. In addition, most of the figures do not refer to the actual effect but to variations of effective refractive index and prior results.

Reply: Thank you for the comment. We have checked the presentation of both the formulas and figures carefully, making sure that the word descriptions are more professional, as well as the actual effect (lines 150-156; lines 164-165; lines 168-169). Because the energy loss in the waveguide can be characterized by the imaginary part of the effective refractive index of graphene, the variations of the effective refractive index are discussed in detail.

(B)A meticulous optimization of the reported effect should be executed for various chemical potentials, scattering times of electrons and background permittivity contrast. The regimes for maximum transmission efficiency should be determined.

Reply: Thanks our reviewer for the professional comments.

For the first point, figure 3 in the paper shows the reported effect versus the Fermi level Ef, the dielectric constant of substrate and the excited wavelength. The chemical potential is proportional to the Fermi level Ef. In addition, the electron scattering time of graphene is a complex and variable physical quantity, we are not sure that whether it would or how influence the maximum efficiency. However, this parameter is not used in the theoretical analysis present in the paper.

   For the second point, the proposed beam splitter is based on self-imaging of the interference of the graphene surface plasmon in the waveguide. For the model, when the refractive index of the graphene is larger than those of the upper and bottom dielectrics, the GSP would reflect on the interface between the graphene and the dielectric, resulting in the multimode interference (MMI) of GSP on graphene. If the exit port is set in the imaging position, the GSP energy would be divide into several parts. Based on the above mentioned splitting principle, the size of the waveguide is changed with the imaging position z of the divided GSP energy. The z can be expressed by the presented equations 11 and 12 in the paper .

;     (equations 11 and 12)

Hence, for different parameters such as chemical potentials or background permittivity, the splitting effect is different, thus the maximum transmission efficiency can’t be contrasted.

(C) The authors should discuss extensively in their revised version the possibility of changing the environment into graphene strips are embedded. How the reported effects are expected to change if the surrounding is cylindrical [1] or the whole graphene structure is finite [2]?

Reply: This is good point. The proposed beam splitter in the paper is based on self-imaging of the interference of the graphene surface plasmon in the waveguide. If we change the environment into other structures such as graphene strips or finite graphene (or cylindrical structure), the graphene surface plasmons possibly would not interfere. To make sure the multimodes interference, we should carefully design the structure such as cylindrical dielectric loaded on the finite graphene, basing on the theory of the guided-mode propagation analysis.

(D) A better novelty statement is required. What is the new that this study can offer to related research domain. The authors should elaborate and compare with competing designs. [1] On examining the influence of a thin dielectric strip posed across the diameter of a penetrable radiating cylinder, Progress In Electromagnetics Research C, 2008. [2] Excitation of guided waves of grounded dielectric slab by a THz plane wave scattered from finite number of embedded graphene strips: Singular integral equation analysis, IET Microwaves, Antennas & Propagation, 2021.

 Reply: Thank you for the comment. To better describe the novelty, we checked again the manuscript. The novelty can be found in the paper (lines 188-189; lines 193-194; lines 198-199). The competing designs were also discussed and compared (lines 171-174). The latter mentioned reference was cited in the paper, as shown in reference 27.

This manuscript is a resubmission of an earlier submission. The following is a list of the peer review reports and author responses from that submission.

Round 1

Reviewer 1 Report

Comments and Suggestions for Authors

The manuscript presents a theoretical analysis of a 1x2 beam splitter on a graphene surface plasmon waveguide, by using according to the authors a self imaging  method.

The work could be of interest and has an indirect but  possibly adequate connection to the Journal Nanomaterials, as several design parameters could be related to material properties and therefore to an optimization process through proper material based selection.

However, the manuscript seriously lacks of basic required key elements of a solid journal paper such as  clarity, organization, discussion, and support of conclusions. The manuscript does not even have separate sections and is organized in a unified text, by not separating the introduction, methodology, discussion etc.

Moreover, the development of the theoretical approach, starting from page 3 is so disorganized, with missing information and assumptions that is impossible to follow.

Furthermore, critical English mistakes need to be corrected

Overall, despite the scientific interest in this area, the manuscript is of such low presentation quality that cannot be considered for publication, and needs drastic rewriting before resubmission.

Comments on the Quality of English Language

 Generally the quality of  English is acceptable.

However there are some scattered serious mistakes, as in the following sentences, that authors need to correct and also proof read the manuscript for other mistakes

 1) Recently, people have realized many graphene plasmonic devices such as tunable

 2)  modulation abilities

 3) The simulation results show that the beam splitting effect is good

Reviewer 2 Report

Comments and Suggestions for Authors
